# Adenosine A$_{2A}$ Receptor Activation Regulates Niemann–Pick C1 Expression and Localization in Macrophages

**Adrienn Skopál** [1,2], **Gyula Ujlaki** [1,2], **Attila Tibor Gerencsér** [1], **Csaba Bankó** [3,4], **Zsolt Bacsó** [3], **Francisco Ciruela** [5,6], **László Virág** [1,7], **György Haskó** [8] and **Endre Kókai** [1,9,*]

1   Department of Medical Chemistry, Faculty of Medicine, University of Debrecen, H-4032 Debrecen, Hungary
2   Doctoral School of Molecular Medicine, University of Debrecen, H-4032 Debrecen, Hungary
3   Department of Biophysics and Cell Biology, Faculty of Medicine, University of Debrecen, H-4032 Debrecen, Hungary
4   Doctoral School of Molecular Cell and Immune Biology, University of Debrecen, H-4032 Debrecen, Hungary
5   Pharmacology Unit, Department of Pathology and Experimental Therapeutics, School of Medicine and Health Sciences, Institute of Neurosciences, University of Barcelona, L'Hospitalet de Llobregat, 08907 Barcelona, Spain
6   Neuropharmacology and Pain Group, Neuroscience Program, Bellvitge Institute for Biomedical Research, L'Hospitalet de Llobregat, 08907 Barcelona, Spain
7   MTA-DE Cell Biology and Signaling Research Group, University of Debrecen, H-4032 Debrecen, Hungary
8   Department of Anesthesiology, Columbia University, New York, NY 10032, USA
9   Section of Dental Biochemistry, Department of Basic Medical Sciences, Faculty of Dentistry, University of Debrecen, H-4032 Debrecen, Hungary
*   Correspondence: ekokai@med.unideb.hu

**Abstract:** Adenosine plays an important role in modulating immune cell function, particularly T cells and myeloid cells, such as macrophages and dendritic cells. Cell surface adenosine A$_{2A}$ receptors (A$_{2A}$R) regulate the production of pro-inflammatory cytokines and chemokines, as well as the proliferation, differentiation, and migration of immune cells. In the present study, we expanded the A$_{2A}$R interactome and provided evidence for the interaction between the receptor and the Niemann–Pick type C intracellular cholesterol transporter 1 (NPC1) protein. The NPC1 protein was identified to interact with the C-terminal tail of A$_{2A}$R in RAW 264.7 and IPMϕ cells by two independent and parallel proteomic approaches. The interaction between the NPC1 protein and the full-length A$_{2A}$R was further validated in HEK-293 cells that permanently express the receptor and RAW264.7 cells that endogenously express A$_{2A}$R. A$_{2A}$R activation reduces the expression of NPC1 mRNA and protein density in LPS-activated mouse IPMϕ cells. Additionally, stimulation of A$_{2A}$R negatively regulates the cell surface expression of NPC1 in LPS-stimulated macrophages. Furthermore, stimulation of A$_{2A}$R also altered the density of lysosome-associated membrane protein 2 (LAMP2) and early endosome antigen 1 (EEA1), two endosomal markers associated with the NPC1 protein. Collectively, these results suggested a putative A$_{2A}$R-mediated regulation of NPC1 protein function in macrophages, potentially relevant for the Niemann–Pick type C disease when mutations in NPC1 protein result in the accumulation of cholesterol and other lipids in lysosomes.

**Keywords:** adenosine receptor; Niemann–Pick C1 protein; lysosome-associated membrane protein 2; protein interaction; macrophage

## 1. Introduction

Adenosine, a naturally occurring purine nucleoside, exerts its physiological effects through G protein-coupled adenosine receptors [1]. The adenosine A$_{2A}$ receptor (A$_{2A}$R) regulates important physiological processes within the cardiovascular, nervous, and immune systems [2]. Specifically, stimulation of A$_{2A}$R in immune cells promotes the release of anti-inflammatory mediators and reduces the production of pro-inflammatory cytokines [3].

Adenosine-mediated activation of cell surface $A_{2A}R$ within immune cells, including neutrophils, regulatory T cells, cytotoxic T cells, macrophages, and dendritic cells [4], promotes receptor coupling with the Gs/olf protein. Subsequently, stimulation of adenylate cyclase triggers the generation of cAMP and the activation of protein kinase A (PKA), which in turn phosphorylates ion channels, transcription factors, enzymes, structural proteins, and signaling proteins [5]. In addition to G-protein coupling, the formation of receptor oligomers and interactions with scaffold proteins are essential for the proper receptor's signaling [6]. Therefore, protein-protein interactions involving $A_{2A}R$ play a key role in its function, as they are critical to determining the overall activity of the receptor and the effectiveness of its signaling response [7].

Currently, a growing number of $A_{2A}R$-interacting proteins have been identified [7,8]. Therefore, the study of the $A_{2A}R$ interactome has shed light on the intricate relationships between the receptor and its interacting partners, leading to a more comprehensive understanding of its function and role in disease processes. Interestingly, the formation of $A_{2A}R$-effector macromolecular membrane assemblies playing a significant role in the regulation of receptor function has been established [6]. This contention leads researchers to explore the potential use of these $A_{2A}R$-containing complexes as a putative therapeutic target for various diseases, such as Parkinson's disease (PD), where modulation of the dopamine $D_2$ receptor (i.e., $D_2R$) by $A_{2A}R$ has been largely studied [9–12]. Specifically, an $A_{2A}R$ antagonist (i.e., istradefylline) has been approved and commercialized under the name of Nourianz® as an 'add-on' treatment for levodopa in PD patients with "off" episodes [13,14]. Furthermore, the $D_2R$-interacting protein GPR37, an orphan G protein-coupled receptor, has also been shown to interact with $A_{2A}R$ [14]. Dopamine/adenosine-mediated functional psychomotor studies in GPR37$^{-/-}$ mice suggested that GPR37 plays a role in modulating $D_2R$-mediated neurotransmission through its association with $A_{2A}R$ [15]. $A_{2A}R$ blockade has shown similar positive results in the experimental treatment of depression and schizophrenia, which beneficial effect is mediated by indirect enhancement of NMDA receptor function [16,17].

Finally, $A_{2A}Rs$ have been linked to the severity of neuropsychiatric diseases by heterodimerization with $A_{1A}Rs$ [18,19] and their usefulness as a potential therapeutic target for the treatment of Alzheimer's disease and Huntington's disease [20,21], as well as chronic stress and memory impairment [22], has been postulated.

Given that immune cells express high levels of $A_{2A}Rs$, we recently set out to identify $A_{2A}R$-interacting partners in macrophages. Our protein-protein interaction mapping identified Cathepsin D as an $A_{2A}R$ interacting protein, the first such $A_{2A}R$-interacting protein to be identified in the macrophages [23]. Interestingly, stimulation of $A_{2A}R$ in macrophages prevents the release of cytotoxic and proinflammatory mediators, leading to a reduction in tissue damage [24–29]. Consistent with this, $A_{2A}R$ activation alleviates symptoms of inflammatory diseases, such as multiple sclerosis (MS) [30], rheumatoid arthritis (RA) [31], acute lung injury (ALI) [32], ischemia-reperfusion injury (IRI) of the kidney [33,34], and inflammatory bowel disease (IBD) [35]. Here, our objective was to expand the $A_{2A}R$ interactome, specifically in macrophages, to better understand how this receptor modulates immune responses in these cells. To do this, we performed co-immunoprecipitation in RAW264.7 and pull-down experiments IPMφ macrophage cells using $A_{2A}R$ as bait. Although several preys involved in the regulation of vesicular transport were found, the Niemann–Pick C1 protein (NPC1) attracted our attention because the regulatory role of $A_{2A}R$ in the rescue of the phenotype of the NPC mutant has previously been demonstrated [36,37] in fibroblasts from Niemann–Pick C1 patients. Cholesterol export requires NPC intracellular cholesterol transporter 1 (NPC1) and NPC2, genetic mutations of which can cause Niemann–Pick type C (NPC) disease, a disorder characterized by massive lysosomal accumulation of cholesterol and glycosphingolipids [38] NPC disease affects various organs and systems in the body, including the liver, spleen, and brain, leading to a range of symptoms that can include hepatosplenomegaly, progressive neurological deterioration, and cognitive decline. NPC1 is located in the membrane of endosomes and lysosomes and is involved

in intracellular cholesterol transport. It plays a role in the transport of cholesterol and other types of fats across cell membranes. When this system is disrupted, cholesterol is deposited and can lead to lysosomal storage diseases, mainly associated with neurological symptoms [39].

## 2. Materials and Methods

### 2.1. Reagents

Materials were ordered from Sigma-Aldrich (St. Louis, MO, USA), except for RAW 264.7 and HEK-293 cell lines (ATCC, New York, NY, USA), HEK-293 transgenic cell line expressing Flag-$A_{2A}R$ (provided by Francesco Ciruela, Department of Pathology and Experimental Therapeutics, University of Barcelona, Spain), DMEM (LM-D1111, GENTAUR Europe BVBA, Kampenhout, Belgium), FBS (FB-1090, GENTAUR Europe BVBA, Kampenhout, Belgium), T75 cell culture dish (Z707546, TPP, Trasadingen, Switzerland), Falcon Multiwell tissue culture plates (351146, BD Biosciences, San Jose, CA, USA), TRI reagent (TR118, Molecular Research Center, Cincinnati, OH, USA), Maxima SYBR Green/ROX qPCR Master Mix (2X) (K0222, Thermo Scientific, Waltham, MN, USA), SDS-PAGE (4568034, Bio-RAD, Hercules, CA, USA), West Pico Super Signal ECL Western Blotting Detection Reagent (34580, Thermo Fisher, Waltham, MA, USA), 8 wells tissue culture treated glass chamber slides (354118, Falcon, Chicago, IL, USA), 96-well Cell Carrier Ultra plate (6055302, Perkin Elmer, Waltham, MA, USA), CGS21680 (1063, Tocris, Minneapolis, MN, USA), phenylmethylsulfonyl fluoride (PMSF-RO, Merck, Darmstadt, Germany), protease inhibitor cocktail (M221, VWR International, Radnor, PA, USA).

### 2.2. Isolation of Mouse Peritoneal Macrophages

Peritoneal macrophage isolation from mouse strain C57BL6/J was performed according to Skopal et al., 2022 [23].

### 2.3. Animal Models

The experiments were carried out with 8–12 week-old C57BL6/J wild-type ($A_{2A}R^{+/+}$) mouse colonies (The Jackson Laboratory, Farmington, CT, USA). All mice were maintained in specific pathogen-free conditions in the Central Animal Facility, and all animal experiments were conducted according to the guidelines of the Declaration of Helsinki and approved by the Institutional Review Board of the University of Debrecen (DEMÁB).

### 2.4. Cell Culture

RAW 264.7 macrophages, HEK-293 cell lines (ATCC, New York, NY, USA) expressing Flag-$A_{2A}R$ (provided by Francesco Ciruela, Department of Pathology and Experimental Therapeutics, University of Barcelona, Spain), and primary IPMϕ were cultured under the conditions described in Skopál et al., 2022 [23].

### 2.5. Immunoprecipitation of cMyc-$A_{2A}R^{284-410}$

The immunoprecipitation experiment was performed as described in our previous publication [23] with the following modifications: 500 µg of protein extract from cMyc-$A_{2A}R^{284-410}$ expressing RAW 264.7 cells in RIPA buffer were diluted with PBS and supplemented with 100× Protease Inhibitor Cocktail (PIC, M221, VWR International, Radnor, PA, USA) and 100× Phenylmethylsulfonyl fluoride (PMSF) to 500 µL. 1 µg anti-cMyc antibody and its isotype control IgG1 (M5546, I5006, Sigma-Aldrich, St. Louis, MO, USA) were added to the lysates and incubated overnight with rotation at 4 °C. The total band protein content of the immuno-complexes formed with anti-cMyc antibody, were separated by SDS-PAGE and visualized by Coomassie Brillant Blue G250 staining, and the signal was detected with Fluorochem FC2 Imaging System (Alpha Innotech, Midland, ON, Canada). The whole band protein content was digested with trypsin and analyzed by mass spectrometry.

*2.6. Purification of GST-A$_{2A}$R$^{284–410}$ Specific Pull-Down Complex*

The GST and GST-A$_{2A}$R$^{284–410}$ recombinant proteins were expressed in the E. coli BLR strain and extracted from the supernatant using a MagneGST purification kit, according to Skopal et al. [23]. Then 20 μg MagneGST bounded GST and GST-A$_{2A}$R$^{284–410}$ recombinant proteins were incubated with 500 μg of mouse peritoneal macrophage (IPMφ) cell lysates overnight with rotation at 4 °C. Bound proteins were eluted from the pull-down complexes with 48 μL ice-cold RIPA buffer containing 12 μL 5× SDS sample buffer. The eluted samples were denatured for 10 min at 95 °C, separated by 10% SDS-PAGE. The separated proteins were visualized by Coomassie Brillant Blue G250 staining, and the signal was detected with Fluorochem FC2 Imaging System (Alpha Innotech, Midland, ON, Canada). The whole band protein content was digested with trypsin and analyzed by mass spectrometry.

*2.7. Mass Spectrometry*

The tryptic fragments of cMyc-A$_{2A}$R$^{284–410}$ and its isotype control specific immunocomplexes from RAW264.7 cell lysates and the tryptic fragments of GST and GST-A$_{2A}$R$^{284–410}$ specific pull-down complexes from mouse IPMφ were analyzed by matrix-assisted laser desorption ionization time-of-flight mass spectrometry in Mass Spectrometry Laboratory of Rutgers University, Newark, NJ as in [40]. The amino acid sequence for one peptide was also confirmed by photothermal deflection spectrometry.

*2.8. Pharmacological Treatment of Macrophages*

The pharmacological reagents were dissolved in water at the following concentrations: A$_{2A}$R agonist (CGS21680 (1063, Tocris, Minneapolis, MN, USA)) 50 mM; LPS (L-3880, 0124:B8, Sigma, St. Louis, MO, USA) 5 mg/mL, respectively. A$_{2A}$R agonist was used in the final concentration of 100 nM, and LPS was utilized in a concentration of 100 ng/mL in the culture media. We pre-incubated the macrophages in the presence of an A$_{2A}$R agonist (CGS21680) for 20 min before LPS was added for 4 h to the media. After the treatment, the total RNA was isolated with TRI reagent, reverse transcribed and used for qRT-PCR as a template. Fold changes were analyzed by the q-RT-PCR method. After the protein isolation, the protein concentration was determined from the cell lysate and used for IP, pull down and immunostaining methods. The immunostained pictures were analyzed by confocal microscopy.

*2.9. Protein Isolation*

After treatment with the different drugs, protein isolation from IPMφ cells was performed according to the method described by Skopál et al 2022 [23]. Protein concentration was determined by Direct Detect Spectrophotometer (Merck-Millipore, Darmstadt, Germany) and Bicinchoninic acid (BCA) Protein Assay Kit (23,225, Thermo Fisher, Waltham, MA, USA). A standard curve was prepared by plotting the OD value at a wavelength of 562 nm corrected to the value of the blank sample for each BSA standard. Then, the standard curve was used to determine the protein concentration of each sample.

*2.10. RNA Extraction and q-RT-PCR*

RNA was purified with TRI reagent as described in the manufacturer's instructions. 2 μg of the cDNA was added to 8 μL of reaction Maxima SYBR Green/ROX qPCR Master Mix (2X). Quantitative real-time PCR was performed under the following conditions: at 95 °C for 10 min, followed by 50 cycles of 94 °C for 10 s, 60 °C for 10 s, and 72 °C for 10 s. Reactions were carried out in triplicate, and data were normalized to the geometric mean of housekeeping genes (β2M, GAPDH). Sequences of primers are given in Table 1. β2M, GAPDH, and NPC1 primers were ordered from IDT (Coralville, IA, USA). Quantitative real-time PCR was performed with a Roche LightCycler 480 II (Roche, Basel, Switzerland).

**Table 1.** Oligoes were used for the q-RT-PCR experiments.

| Primers | Forward | Reverse |
|---------|---------|---------|
| β2M | 5′-AGTATACTCACGCCACCCAC-3′ | 5′-CATGTCTCGATCCCAGTAGACG-3′ |
| NPC1 | 5′-TTTGGTATGGAGAGTGTGGA-3′ | 5′-ACAGCAGAGACTGACATTGT-3′ |
| GAPDH | 5′-ACAGTCCATGCCATCACTG-3′ | 5′-GCCTGCTTCACCACCTTCTT-3′ |

## 2.11. Immunoprecipitation of cMyc-$A_{2A}R^{284-410}$ and NPC1 Proteins

Immunoprecipitation of cMyc-$A_{2A}R^{284-410}$ and NPC1 Proteins from RAW 264.7 and HEK cell lysates containing 500 µg of protein extract was performed as described in our previous publication [23] with the following modifications: 1 µg anti-cMyc antibody and its isotype control IgG1 (Table 2. M5546, I5006, Sigma-Aldrich, St. Louis, MO, USA), 8.5 µg anti-$A_{2A}R$ antibody or 8.5 µg control rabbit serum (Sigma-Aldrich, St. Louis, MO, USA) were added to the lysates. The eluted samples were analyzed by WB using an anti-cMyc or anti-NPC1 antibody (Table 2).

**Table 2.** Antibodies used for western blotting (WB), immunostaining (IS), nuclear and F-actin staining dye were used for IS.

| Antibody | Methods | Applied Concentration | Catalog Number; Supplier |
|----------|---------|----------------------|--------------------------|
| anti-$A_{2A}R$ | IP | 4.25 µg/mL | AAR-002; Alomone labs (Jerusalem, Israel) |
| anti-cMyc | WB<br>IP | 1.67 µg/mL<br>16.5 µg/mL | M5546, Sigma-Aldrich, (Budapest, Hungary) |
| anti-EEA1 | IS | 1 µg/mL | SAB4300682; Sigma Aldrich (Budapest, Hungary) |
| anti-NPC1 | WB<br>IS | 1 µg/mL<br>5 µg/mL | NB400-148, Novus Biologicals (Centennial, CO, USA) |
| anti-rabbit-HRP | WB | 0.2 µg/mL | 7074S, Cell Signaling Technology (Danvers, MA, USA) |
| anti-Mouse-HRP | WB | 0.2 µg/mL | 7076S; Cell Signaling Technology (Danvers, MA, USA) |
| anti-β-Actin-HRP | WB | 0.1 µg/mL | sc-47778 HRP, Santa Cruz Biotechnology (Dallas, TX, USA) |
| anti-Rabbit-Alexa-488 | IS | 5 µg/mL | A27034, ThermoFisher (Waltham, MA, USA) |
| Anti-LAMP2-Alexa-488 | IS | 5 µg/mL | 108510, BioLegend (San Diego, CA, USA) |
| DAPI | IS | 20 µg/mL | D1306, Thermo Fisher (Waltham, MA, USA) |
| Texas Red-X Phalloidin | IS | 5 µg/mL | T7472, Thermo Fisher (Waltham, MA, USA) |

## 2.12. Immunoblot

Ten µg of protein lysates of each sample were denatured for 10 min at 95 °C in the presence of SDS sample buffer and separated on 10% SDS-PAGE (BioRad, Laboratories, Hercules, CA, USA) at 100 V for 60 min. The separated proteins were transferred to a nitrocellulose membrane (10600016, Sigma-Aldrich, Budapest, Hungary) at 400 mA for 90 min. After blocking with 3% BSA in 1× TBST buffer, membranes were incubated with NPC1 specific antibody overnight at 4 °C. The following day, membranes were incubated with anti-rabbit-HRP and anti-β-Actin-HRP antibodies (Table 2) for 1 h at room temperature. Bands were detected using the ECL Western Blotting Detection Reagent (34580, Super Signal, West Pico, Thermo Fisher, Waltham, MA, USA). The signal was detected with Chemidoc Touch Imaging System (Bio-Rad Laboratories, Hercules, CA, USA)

and quantified by Image Lab (Bio-Rad Laboratories, Hercules, CA, USA) and Image J software (version 4/2020).

### 2.13. Immunostaining of LAMP2, EEA1 and NPC1 Protein

IPM$\phi$ (3 × 105) and RAW264.7 (5 × 104) cells were cultured in 300 μL DMEM in 8 wells tissue culture treated glass chamber slides (354,118, Falcon, Chicago, IL, USA) and IPM$\phi$ (105) and RAW264.7 (2 × 104) cells 96 well Cell Carrier Ultra plates (6,055,302, Perkin Elmer, Waltham, MA, USA). Cells were pre-treated with A$_{2A}$R agonist (CGS21680, 100 nM) for 20 min before LPS (100 ng/mL) was added for 4 h to the cells. After the treatment, the cell culturing media was changed to fresh DMEM. The detection of LAMP2 protein was performed in live cell staining, the anti-LAMP2-Alexa-488 antibody (Table 2) was added to the cell culturing media in the concentration of 5 μg/mL and cells were further incubated for 30 min at 37 °C. Then cells were fixed with 4 *w/v*% PFA solution for 20 min and were incubated in a blocking buffer (2 *w/v*% of BSA dissolved in PBS) at room temperature for 30 min. In EEA1 and NPC1 specific IS, the cells were fixed with 4 *w/v*% PFA solution for 20 min and were incubated in blocking buffer (2 *w/v*% of BSA dissolved in PBS) at room temperature for 30 min. Anti-EEA1 (Table 2) and anti-NPC1 antibody (Table 2) were added in the concentration of 1 μg/mL and 5 μg/mL and cells were incubated overnight at 4 °C. Cells were washed three times with 300 μL PBS, then Alexa-488-conjugated anti-rabbit secondary antibody (Table 2) for EEA1 and NPC1 staining was added in a concentration of 5 μg/mL to the blocking buffer and incubated for 1 h at room temperature. Nuclei were stained with 20 μg/mL DAPI (Table 2) for 1 h at room temperature in the blocking buffer. After staining, chamber slides were covered with 5 μL of Mowiol-Dabco mounting medium (81381, Sigma-Aldrich, St. Louis, MO, USA) and coverslips (24 × 60 mm, Hirschmann Laborgerate, Eberstadt, Germany). Photos were taken by Leica SP8 confocal microscope (Leica Microsystems, Buffalo Grove, IL, USA) using a 63× oil immersion objective (NA: 1.4).

In the 96-well Cell Carrier Ultra plate, the blocking buffer was changed to 50 μL PBS and images were acquired by Opera Phenix High Content Confocal System (Perkin Elmer, Waltham, MA, USA). 50–212 fields and 300–6100 cells were acquired per well, and laser-based autofocus was performed at each imaging position. Images of DAPI and Alexa-488 channels were collected at 2 μm of the Z image plane using a 63× water immersion objective (NA: 1.15) to visualize the cells and the localization of A$_{2A}$R. The primary data were analyzed by Harmony 4.8 software (Perkin Elmer, Waltham, MA, USA) according to the Spot Analyses Ready to Made Solution (http://www.perkinelmer.com/product/harmony-4-2-office-hh17000001; accessed on 1 January 2019) with custom modifications.

At the beginning of our immunostaining experiments, we used actin-specific staining (Texas Red-X Phalloidin, Thermo Fischer, Waltham, MA USA) to verify the accuracy of Harmony4.8 software for the High Content Confocal System to identify the outer boundary of macrophage cytoplasm based on specific staining of the proteins of interest (NPC1, EEA1). The results of two independent immunostaining experiments for NPC1 and EEA1 (Figures S5 and S6, respectively, see Supplementary Materials) showed that the outer boundary of the cytoplasm identified by Harmony4.8 overlapped with the outer boundary of the cytoplasm labeled by actin staining, so the specific NPC1, EEA1, and LAMP2 labels were used in the remaining independent experiments.

Image intensities were rescaled, then cells were identified on the DAPI signal, and the cellular phenotypes were characterized on the basis of the Alexa488 signal. Cellular features, such as the number of spots, total spot area, and relative spot intensity in membrane and cytoplasm regions, were extracted. The statistical analyses of the parallel data set were made by GraphPad Prism 8 program. The evaluation of the data based on the individual analysis of 300–6100 presented as mean ± SEM. * $p < 0.05$, ** $p < 0.01$ and *** $p < 0.001$ vs. control (vehicle-treated); # $p < 0.05$, ## $p < 0.01$ and ### $p < 0.001$ vs. LPS-treated cells.

*2.14. Laser Scanning Cytometry*

105/well RAW 264.7 cells were labeled with anti-LAMP2-Alexa-488 antibody (Table 2) in the concentration of 0.5 mg/mL (green), and the nucleus was stained with DAPI (Table 2) for 1 h at room temperature in a concentration of 20 µg/mL in the blocking buffer. Samples were measured with a slide-based laser-scanning iCys Research Imaging Cytometer (Thorlabs Imaging Systems, Sterling, VA, USA). The arising fluorescence signals (405 nm and 488 nm) were collected by a 40× (NA 0.75) objective into 2 detection channels (blue and green channels). PMT settings for the blue channel were 22 V, the gain was 100%, and the offset was −0.14 V. In the green channel, PMT was set to 40 V, the gain to 100%, and the offset to −0.06 V. The X-step size was 0.25 µm. The field size was 250 × 192 µm. The resolution was 1024 × 768, and the pixel size was 0.25 µm × 0.25 µm.

*2.15. Statistical Analyses*

Data are presented as mean ± SEM of three-six independent experiments. D'Agostino & Pearson tests were used to analyze normality. In one case, when the data show normal distribution, One-way ANOVA was performed, complemented with Sidak's post hoc test. In the other case, if the data did not show normal distribution, the data were transformed then One-way ANOVA was performed, complemented with Sidak's post hoc test. $p$ values < 0.05 were considered statistically significant (* $p < 0.05$; ** $p < 0.01$; *** $p < 0.001$). Statistical analyses were performed with GraphPad Prism 8.0 software (GraphPad Software Inc., San Diego, CA, USA).

## 3. Results

*3.1. Identification of $A_{2A}R$-Interacting Proteins by Co-Immunoprecipitation and Pull-Down Experiments Coupled with Mass Spectrometry Detection*

The C-terminus of mouse $A_{2A}R$ (GenBank Accession No. NM_009630) was used as bait in co-immunoprecipitation (co-IP) and pull-down (PD) experiments. Co-IP was performed using the RAW 264.7 mouse macrophage cell line. For this purpose, the C-terminus of the $A_{2A}R$ tagged with cMyc epitope-tagged $A_{2A}R$ (i.e., cMyc-$A_{2A}R^{284-410}$) was expressed in RAW 264.7 cells (Figure 1A). The cell extracts were then co-immunoprecipitated using a cMyc-specific antibody, and the proteins present in the immune complex formed (Figure 1C) were identified by mass spectrometry. In parallel, a control isotype antibody was used in the co-IP experiment to identify nonspecific interactors. Similarly, to confirm the results obtained in co-IP, PD experiments were performed using the C-terminus of $A_{2A}R$ fused to the GST protein (i.e., GST-$A_{2A}R^{284-410}$) performed in IPMφ extracts (Figure 1B). Additionally, a PD experiment using GST protein alone was run in parallel to identify nonspecific interactors. Importantly, only specific $A_{2A}R$ partners identified both in co-IP and PD experiments were considered (Table 3). Interestingly, one of the proteins identified was cathepsin D, which we had recently reported as an $A_{2A}R$ interacting partner in macrophages using a yeast two-hybrid screen [23], thus validating the approach used here.

**Table 3.** Identified $A_{2A}R$ interacting proteins The $A_{2A}R$ "bait" protein is highlighted by italic proteins that play a role in cellular vesicular trafficking, highlighted by blue.

| Identified Protein | Protein ID | No of Unique Peptide | Identified/Total Aminoacids |
|---|---|---|---|
| Adenosine receptor 2A | Q60613 | 6 | 52/410 |
| Coatomer subunit gamma-2 | Q9QXK3 | 1 | 29/871 |
| Niemann–Pick C1 protein | O35604 | 2 | 24/1278 |
| Isoform 2 of splicing factor 3B subunit 3 | Q921M3–2 | 2 | 27/1122 |
| Sec1 family domain-containing protein 1 | Q8BRF7 | 2 | 34/639 |
| Chaperone protein DnaJ | Q3TK61 | 2 | 48/397 |
| Thyroid hormone receptor-associated protein | Q569Z6 | 1 | 19/951 |
| P2X7 purinoceptor | J7IR93 | 3 | 34/366 |
| Cathepsin D | P18242 | 2 | 36/410 |
| Monoacylglycerol lipase ABHD12n | Q99LR1 | 2 | 23/398 |

**Table 3.** *Cont.*

| Identified Protein | Protein ID | No of Unique Peptide | Identified/Total Aminoacids |
|---|---|---|---|
| AP-3 complex subunit mu-1 | H7BWY2 | 1 | 26/364 |
| Ras-related protein Rab 18 | P35293 | 2 | 28/206 |
| RUN and FYVE domain containing protein | Q8BIJ7 | 1 | 14/712 |
| Myeloid cell nuclear differentiation antigen-like protein | D0QMC3 | 1 | 12/538 |
| Caprin-1 | Q60865 | 1 | 40/707 |
| Myof protein | B9EK95 | 3 | 37/2061 |
| DNA topoisomerase 1 | Q04750 | 3 | 35/767 |
| Alpha glucosidase 2 alpha neutral subunit | A1A4T2 | 2 | 27/966 |
| Receptor mediated endocytosis-8 | D4AFX7 | 1 | 15/2248 |
| T-complex protein 1 subunit gamma | E9Q133 | 2 | 24/507 |
| Heterogeneous nuclear ribonucleoprotein U-like protein 2 | Q00PI9 | 2 | 23/745 |
| Elongation factor Tu | D3YVN7 | 2 | 35/452 |
| DEAD (Asp-Glu-Ala-Asp) box polypeptide 21 | Q6PCP0 | 2 | 37/851 |
| Annexin A5 | P48046 | 2 | 31/319 |
| O-acyltransferase | Q06EZ3 | 1 | 9/540 |
| Serine/threonine-protein phosphatase | Q8BN07 | 1 | 11/285 |
| Histone deacetylase | D3YYI8 | 1 | 12/482 |
| Coronin-1B | Q9WUM3 | 2 | 24/484 |

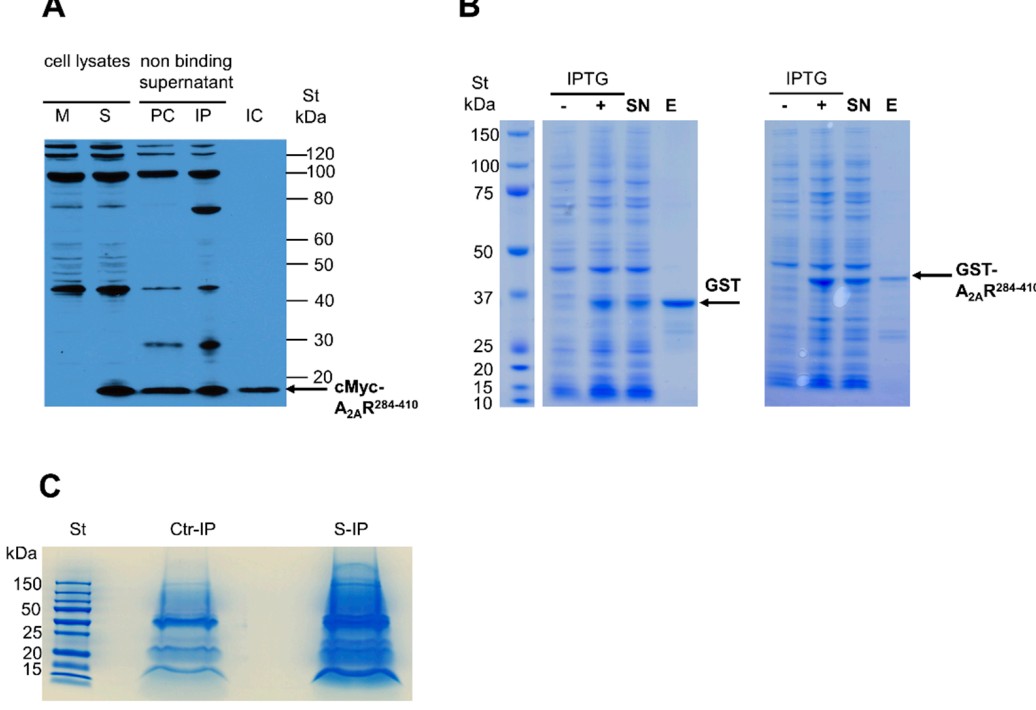

**Figure 1.** Identification of $A_{2A}R^{284-410}$ interacting proteins by co-IP and PD. (**A**) cMyc-$A_{2A}R^{284-410}$ expression and co-IP from RAW 264.7 cells were detected using an anti-cMyc antibody. Lysates from mock (M) and cMyc-$A_{2A}R^{284-410}$ transfected (S) cells. Non-binding supernatants of precleared (PC) and immunoprecipitated (IP) complexes. Eluted co-IP complex (IC). (**B**) Recombinant GST and GST-$A_{2A}R^{284-410}$ proteins were expressed in *E. coli* BLR strain after induction with isopropyl-β-D-thiogalactopyranoside (IPTG) and were affinity purified from the supernatant (SN) fraction of the cell lysates. Recombinant proteins were eluted using 50 mM glutathione and were stained by Coomassie Brillant Blue G250 after separation by SDS-PAGE. (**C**) Samples prepared for mass spectrometry analyses of isotype control (Ctr-IP) and cMyc antibody specific (S-IP) complexes.

List of proteins that were present in both co-IP and PD complexes formed with the C-terminal domain of $A_{2A}R$ ($A_{2A}R^{284-410}$). Proteins interacting with the $A_{2A}R^{284-410}$ protein fragment were identified by mass spectrometry in both experiments (Table 3).

### 3.2. Validation of $A_{2A}R$ and Endogenous NPC1 Interaction in Living Cells

From the identified $A_{2A}R$ interactors (Table 3), we focused on the NPC1 protein, as it has been reported to have a functional interaction with the receptor. To be specific, Popoli and co-workers showed that $A_{2A}R$ activation could restore mitochondrial membrane potential and cholesterol accumulation alterations in NPC1 patient fibroblasts and human neuronal and oligodendroglial NPC1 cell lines, suggesting a potential therapeutic role for $A_{2A}R$ in the treatment of this rare genetic disorder [36,37]. Therefore, we first aimed to validate that the interaction between NPC1 with the C-terminal tail of the receptor ($A_{2A}R^{284-410}$) also occurred with full-length $A_{2A}R$. To this end, we employed HEK-293 cells permanently expressing human $A_{2A}R$ tagged with Flag and SNAP proteins at its N-terminus (i.e., HEK-293-Flag-$A_{2A}R^{SNAP}$ cells; [41]), which we had previously used to demonstrate the interaction of full-length $A_{2A}R$ with cathepsin D [23]. Thus, when $A_{2A}R$ was immunoprecipitated from HEK-293-Flag-$A_{2A}R^{SNAP}$ cell extracts, the NPC1 protein was also found in the resulting immune complexes (Figure 2A). Importantly, the NPC1 protein was not observed when we performed the IP with rabbit control serum (Figure 2A).

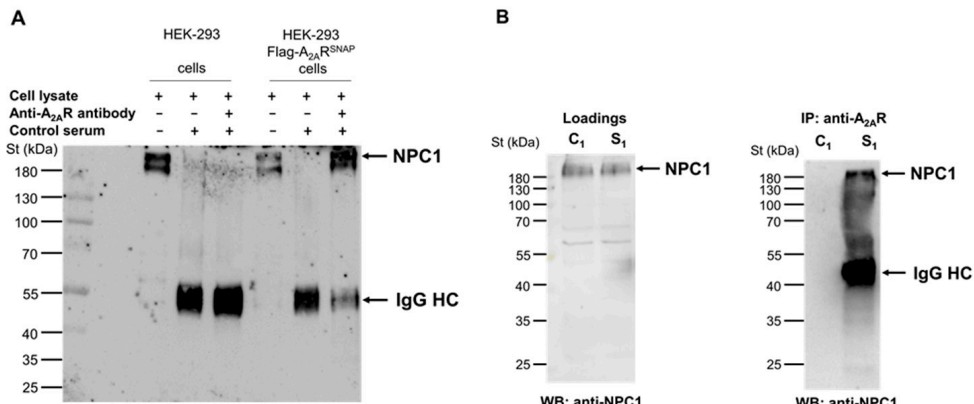

**Figure 2.** Adenosine $A_{2A}$ receptor interacts with NPC1 protein in HEK-293 Flag-$A_{2A}R^{SNAP}$ and RAW264.7 cells. (**A**) HEK-293 and HEK-293 Flag-$A_{2A}R^{SNAP}$ cell lysate (**B**) RAW264.7 cell lysate containing 500 μg protein were incubated with 8.5 μg anti-$A_{2A}R$ antibody. The antibody-containing complex was incubated with Dynabeads Protein G to form the binding. The specific bands were detected in the immuno-complex by western blot using NPC1 specific antibody and anti-rabbit-HRP secondary antibody. In the cell lysates, 5 μg of total proteins were analyzed in each lane. St denotes Prestained Protein Ladder (Thermo Fisher, Waltham, MA, USA). IgG HC: immunoglobulin heavy chain. Data are representative of three independent experiments. The IP was carried out using 8.5 μg of anti-$A_{2A}R$ specific antibody or 8.5 μg control rabbit serum in panel A. The IP was carried out using 8.5 μg of specific antibody in the sample S1. Sample C1 did not contain an anti-$A_{2A}R$ specific antibody in panel B.

Next, we explored whether the interaction between endogenous $A_{2A}R$ and NPC1 also occurs in macrophages. To this end, we performed similar Co-IP experiments in RAW 264.7 cells. Thus, the anti-$A_{2A}R$ antibody was able to co-immunoprecipitate the NPC1 protein from RAW 264.7 cell lysates (Figure 2B), similar to that observed in HEK-293 flag-$A_{2A}R^{SNAP}$ cells (Figure 2A). Importantly, this NPC1-specific protein band was not observed in the absence of an anti-$A_{2A}R$ antibody in the co-IP experiment (Figure 2B). Overall, these results unequivocally demonstrated that full-length $A_{2A}R$ and NPC1 interact not only in HEK-293-Flag-$A_{2A}R^{SNAP}$ cells but also in macrophages.

### 3.3. $A_{2A}R$ Activation Reduces NPC1 mRNA Expression and Protein Density in LPS-Activated Macrophages

We then studied the possibility of a functional interaction between NPC1 and $A_{2A}R$. Therefore, we first examine the impact of $A_{2A}R$ activation on NPC1 mRNA expression and protein density in LPS-activated macrophages. Importantly, when LPS-activated IPMφ cells

were challenged with CGS21680, an $A_{2A}R$ agonist, a significant reduction in NPC1 mRNA relative amount was observed (Figure 3A). Interestingly, $A_{2A}R$ stimulation potentiated LPS-induced down-regulation of NPC1 mRNA relative amount (Figure 3A). Furthermore, when NPC1 protein density was assessed, a significant reduction in NPC1 protein amount was observed in CGS21680 treated IPMϕ cells (Figure 3B). In general, these results suggest that $A_{2A}R$ signaling regulates NPC1 expression in macrophages. Whether this is due to the physical interaction between the two proteins remains to be determined.

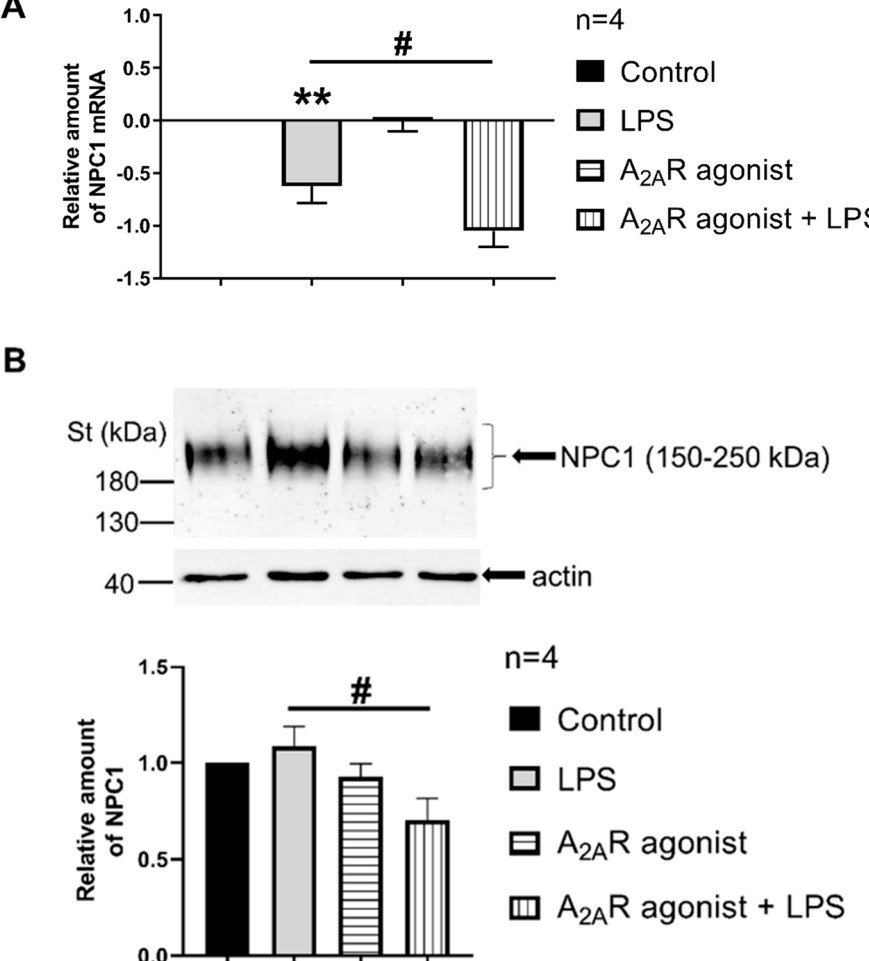

**Figure 3.** $A_{2A}R$ activation decreases NPC1 expression in mouse IPMϕs. (**A**) RNA samples were isolated from IPMϕ cells after LPS activation and treatment with the $A_{2A}R$ agonist (100 nM CGS21680 for 4 h). The reverse transcription was performed with 2 μg of purified total RNA, and Npc1-specific mRNA expression was measured by quantitative RT-PCR. All data were normalized to mouse beta2-microglobulin and GAPDH housekeeping genes. Data are presented as mean ± SEm. $p = 0.4495$ vs. LPS treated cells. (**B**) Protein samples were isolated from mouse IPMϕ cells after the same treatment as in panel A. 10 μg of total protein sample in each lane were analyzed by Western blotting using NPC1-specific polyclonal antibody. Sample loading was normalized for α-actin. Statistical analyses of the relative amount of NPC1 are based on four independent experiments. Statistical analyses are presented as mean ± SEM. # $p < 0.05$ vs. LPS-treated cells; ** $p < 0.01$ vs. untreated cells.

### 3.4. NPC1 Cell Surface Expression in Mouse Macrophages Is Controlled by $A_{2A}R$ Activation

Next, our objective was to determine whether $A_{2A}R$ activation affects the targeting of NPC1 to the cell surface. To this end, we monitored the subcellular distribution of NPC1 in CGS21680-treated macrophages (i.e., RAW 264.7, IPMϕ) in the absence and presence of LPS through high-content confocal microscopy. Interestingly, the activation of RAW 264.7 and

IPMϕ cells with LPS induced an increasing trend, although not significant, in the number of specific spots for the NPC1 protein, the relative intensity of the spot, the total area of the spot, and the number of spots per area in both the plasma membrane and cytoplasmic regions of the cells compared to untreated cells (Figures 4B, 5B and S1A,B). Importantly, CGS21680 pretreatment decreased the number of NPC1-specific spots (Figure 4B) and the area of total spots per membrane region (Figure S1A) in LPS-stimulated RAW264.7. In IPMϕs, CGS21680 decreased the number and total area of NPC1-specific spots detected in the cytoplasm (Figures 5B and S1B) and the intensity of relative NPC1-specific spots identified in the plasma membrane region (Figure S1B). These findings showed that $A_{2A}R$ stimulation decreases the presence of NPC1 protein in the plasma membrane of LPS-activated macrophages.

### 3.5. $A_{2A}R$ Stimulation Decreases Lysosomal-Associated Membrane Protein 2 (LAMP2) Expression in Mouse Macrophages

NPC1 is a genetic disorder that affects the transport of LDL-derived cholesterol from the lumen to the membrane of the lysosome [42]. Interestingly, treatment with an $A_{2A}R$ agonist has been reported to decrease LAMP2, a lysosomal marker [32], in both healthy fibroblasts and fibroblasts from NPC1 patients [37]. Thus, our objective was to investigate the effect of $A_{2A}R$ activation on the intracellular distribution of LAMP2 in resting and activated macrophages. To this end, RAW 264.7 and IPMϕ cells were pretreated with CGS21680, incubated in the absence or presence of LPS, and the cell surface density of LAMP2 were first monitored by laser scanning citometry (Figure S2) and then by high content confocal microscopy, where we determined the number and fluorescence intensity of LAMP2 specific spots in the plasma membrane and cytoplasmic regions of RAW 264.7 cells and IPMϕs. Interestingly, while LPS activation did not alter LAMP2 cell surface density in RAW 264.7 cells (Figures 6B and S3A), it significantly increased the number of LAMP2-specific spots, total spot area and number of spots per area in both plasma membrane and cytoplasmic regions in IPMϕ cells (Figures 7B and S3B). Importantly, pretreatment of RAW 264.7 and IPMϕ cells with CGS21680 precluded LPS-induced subcellular redistribution of LAMP2. In addition, similar changes were observed in the cytoplasmic regions of both types of macrophage cells when pretreated with CGS21680 (Figures 6B, 7B and S3A,B). These results indicate that $A_{2A}R$ activation reduces the amount of LAMP2 protein in LPS-activated macrophages.

### 3.6. $A_{2A}R$ Activation Modulates Early Endosome Antigen 1 (EEA1) Expression in Mouse Macrophages

EEA1, a protein involved in the recycling of early endosomes [39], has been used as a marker for early endosomes during macrophage endocytosis [43]. In CHO cells carrying the NPC1-null mutation, an increase in the size of the endocytotic vesicle was reported when using EEA1 as a marker, suggesting that the NPC1-null mutation may lead to alterations in the endocytic pathway and the formation of larger endocytic vesicles, which may contribute to the cellular dysfunction associated with NPC1. In these enlarged peripheral vesicles, retromer proteins accumulated and endolysosome fusion was inhibited [44]. Therefore, we next examined the effect of $A_{2A}R$ stimulation on the expression of the EEA1 protein in mouse IPMϕs. EEA1 expression and localization were detected by high-throughput confocal microscopy following immunofluorescence labeling and evaluated by the instrument software. Our result showed that LPS treatment of cells increased the total area of the EEA1 spot and the number of spots on the plasma membrane but did not result in a similarly significant change in the cytoplasm. (Figure S4). CGS21680 reduced the number of EEA1-specific spots and the total area of the spot in the plasma membrane, and the total area of the spot in the cytoplasm in cells treated with LPS (Figures 8B and S4).

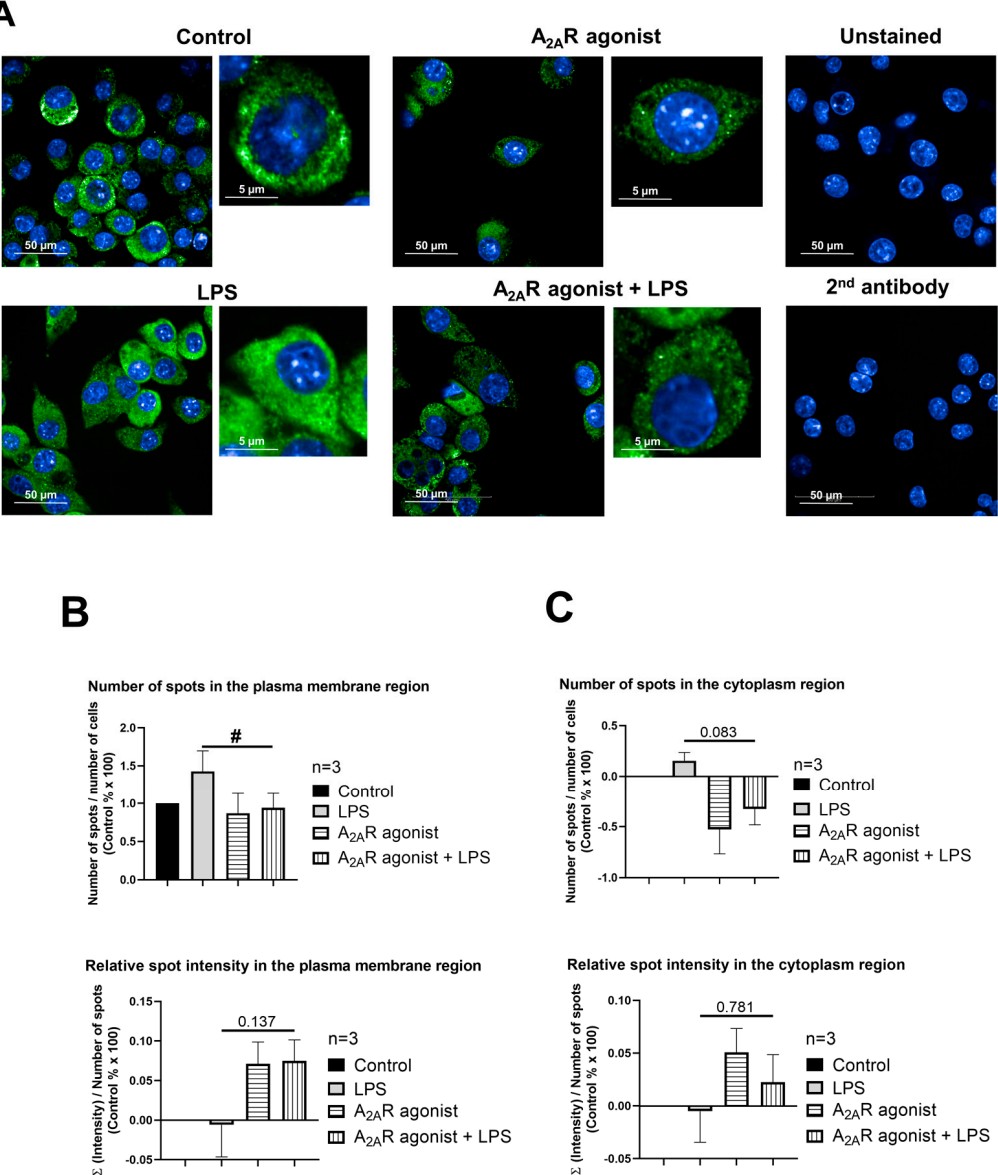

**Figure 4.** $A_{2A}R$ activation decreases cell surface targeting of NPC1 in RAW 264.7 cells. (**A**) Immunofluorescence staining of RAW 264.7 cells was made using NPC1 specific, primary and Alexa-488 conjugated anti-rabbit secondary antibody (green). The nuclei of macrophages were stained with DAPI (blue). NPC1 specific fluorescence intensity was measured after LPS activation and treatment with the $A_{2A}R$ agonist CGS21680 by High Content Analysis, as described in the Section 2. 154–212 fields and 400–6100 cells were acquired per well, and laser-based autofocus was performed at each imaging position. Images of DAPI and Alexa-488 channels were collected at 2 μm of the Z image plane using a 63× water immersion objective (NA: 1.15). Cellular features, such as the number of spots and relative spot intensities in the (**B**) membrane and (**C**) cytoplasmic regions, were extracted. Data obtained from the individual analysis of 400–6100 different cells are presented as mean ± SEM. # $p < 0.05$ LPS vs. LPS + $A_{2A}R$ agonist-treated cells.

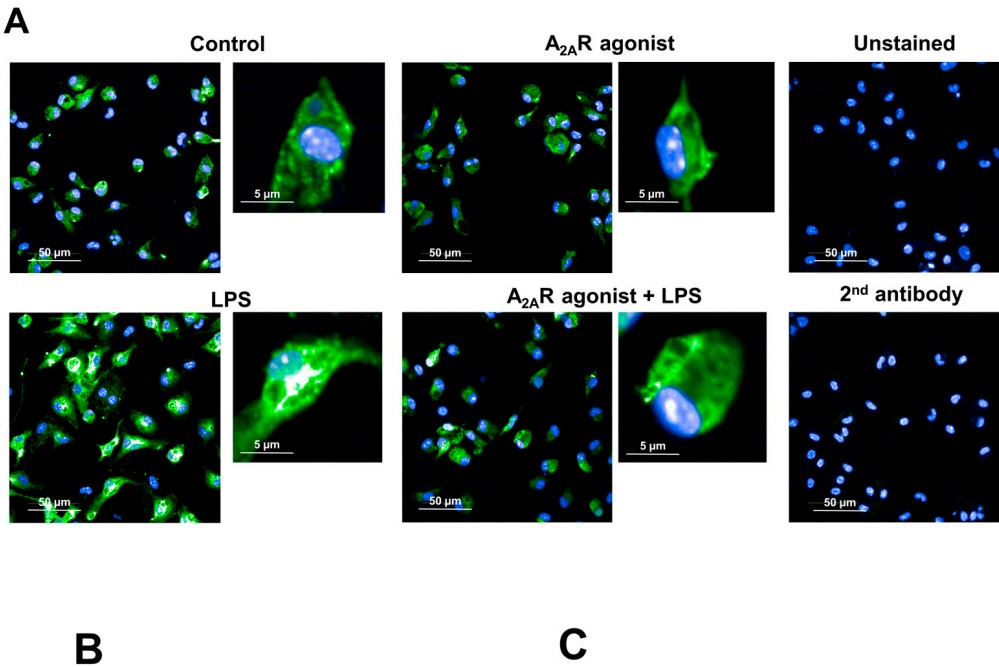

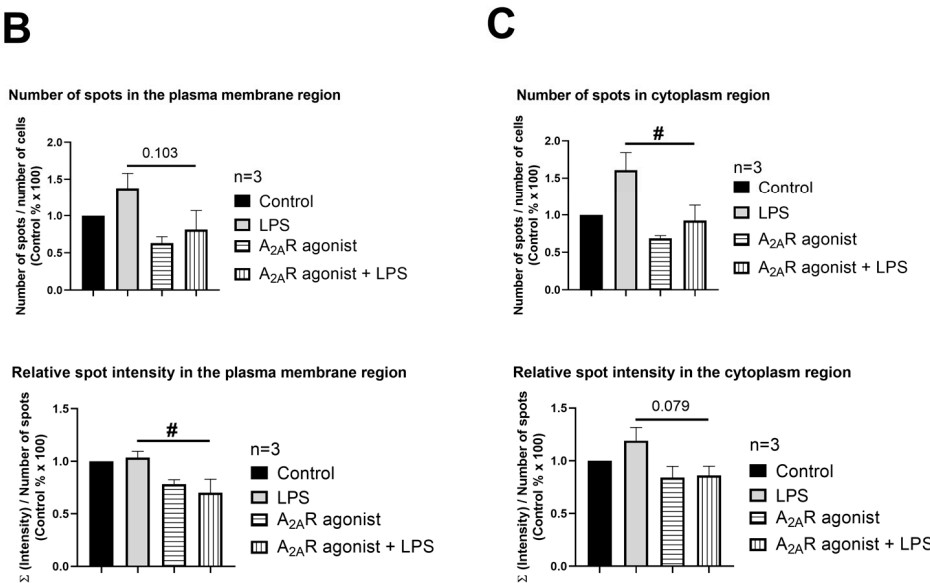

**Figure 5.** $A_{2A}R$ activation decreases cell surface targeting of NPC1 in mouse IPMφs. (**A**) Immunofluorescence staining of mouse IPMφ cells was made using NPC1 specific, primary and Alexa-488 conjugated anti-rabbit secondary antibody (green). The nuclei of macrophages were stained with DAPI (blue). NPC1 specific fluorescence intensity was measured after LPS activation and treatment with the $A_{2A}R$ agonist CGS21680 by High Content Analysis, as described in the Section 2. 66–145 fields and 300–3470 cells were acquired per well, and laser-based autofocus was performed at each imaging position. Images of DAPI and Alexa-488 channels were collected at 2 μm of the Z image plane using a 63× water immersion objective (NA: 1.15). Cellular features, such as the number of spots and relative spot intensities in the (**B**) membrane and (**C**) cytoplasmic regions, were extracted. Data obtained from the individual analysis of 300–3470 different cells are presented as mean ± SEM. # $p < 0.05$ LPS vs. LPS + $A_{2A}R$ agonist treated cells.

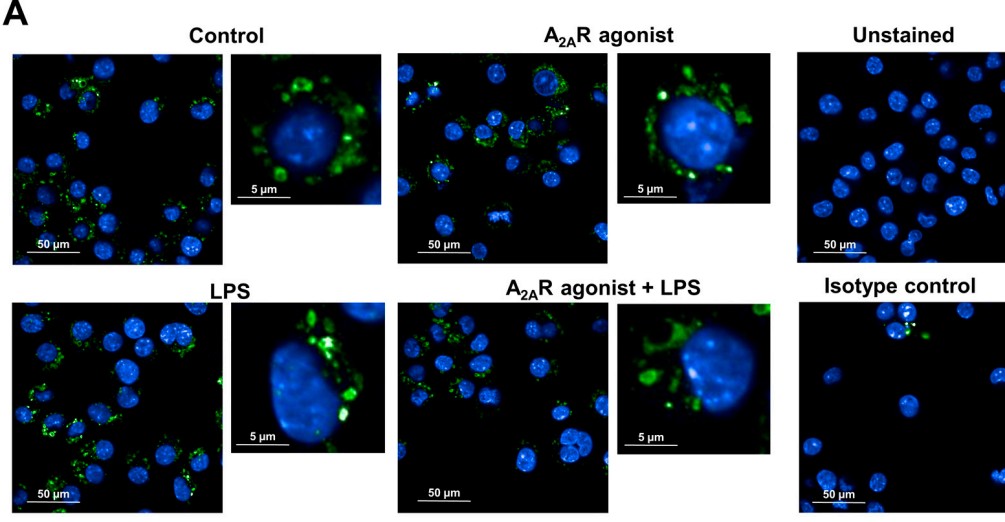

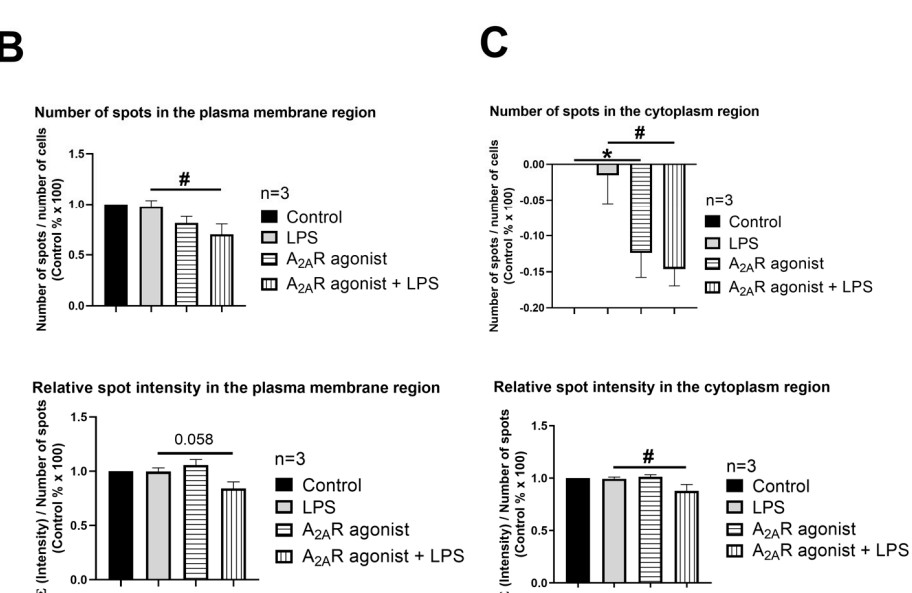

**Figure 6.** A$_{2A}$R activation decreases cell surface targeting of LAMP2 in RAW 264.7 cells. (**A**) Immunofluorescence staining of RAW 264.7 cells was made using LAMP2 specific, Alexa-488 conjugated antibody (green). The nuclei of macrophages were stained with DAPI (blue). LAMP2-specific fluorescence intensity was measured after LPS activation and treatment with the A$_{2A}$R agonist CGS21680 by Opera Phenix High Content Confocal System (Perkin Elmer, Waltham, MA, USA). 154–166 fields and 550–5450 cells were acquired per well, and laser-based autofocus was performed at each imaging position. Images of DAPI and Alexa-488 channels were collected at 2 μm of the Z image plane using a 63× water immersion objective (NA: 1.15). Cellular features, such as the number of spots and relative spot intensities in the (**B**) membrane and (**C**) cytoplasmic regions, were extracted. Data obtained from the individual analysis of 550–5450 different cells are presented as mean ± SEM. * $p < 0.05$ control (vehicle-treated) vs. LPS activated cells; # $p < 0.05$ LPS vs. LPS + A$_{2A}$R agonist-treated cells.

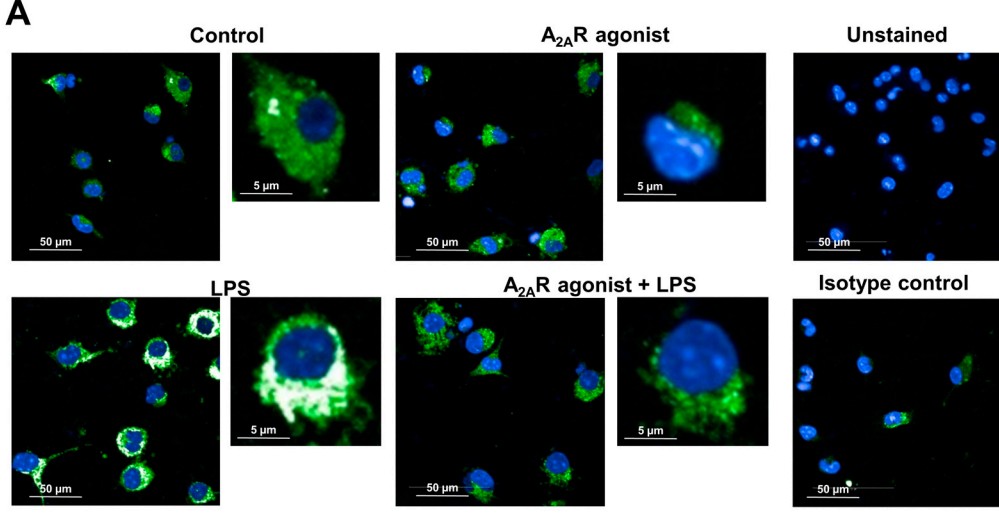

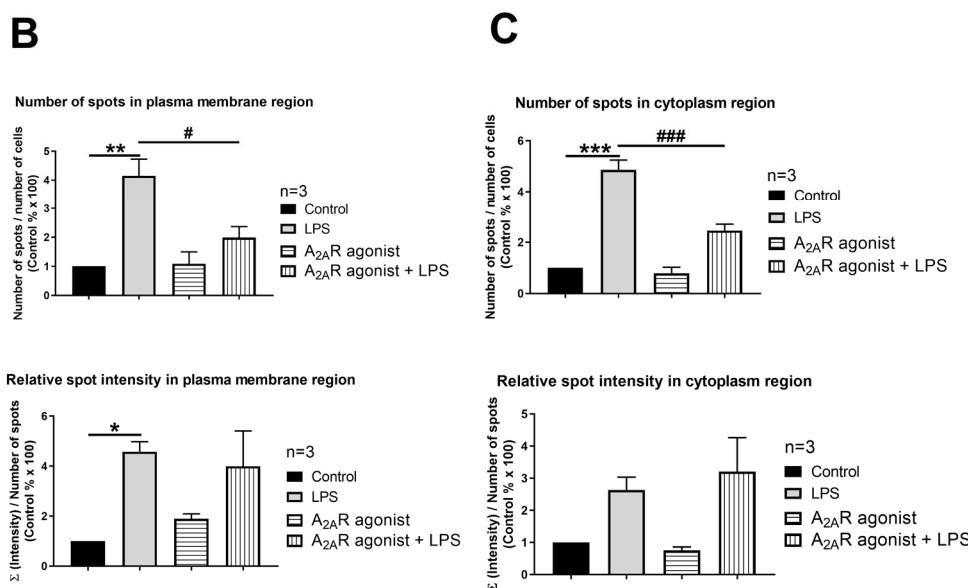

**Figure 7.** A$_{2A}$R activation decreases cell surface deposition of LAMP2 in mouse IPMϕs. (**A**) Immunofluorescence staining of IPMϕ cells was made using LAMP2 specific, Alexa-488 conjugated antibody (green). The nuclei of macrophages were stained with DAPI (blue). LAMP2-specific fluorescence intensity was measured after LPS activation and treatment with the A$_{2A}$R agonist CGS21680 by Opera Phenix High Content Confocal System (Perkin Elmer, Waltham, MA, USA). Fifty fields and 500–1350 cells were acquired per well, and laser-based autofocus was performed at each imaging position. Images of DAPI and Alexa-488 channels were collected at 2 μm of the Z image plane using a 63× water immersion objective (NA: 1.15). Cellular features, such as the number of spots and relative spot intensities in the (**B**) membrane and (**C**) cytoplasmic regions, were extracted. Data obtained from the individual analysis of 500–1350 different cells are presented as mean ± SEM. * $p < 0.05$; ** $p < 0.01$; *** $p < 0.001$ control (vehicle-treated) vs. LPS activated cells and # $p < 0.05$; ### $p < 0.001$ LPS vs. LPS + A$_{2A}$R agonist-treated cells.

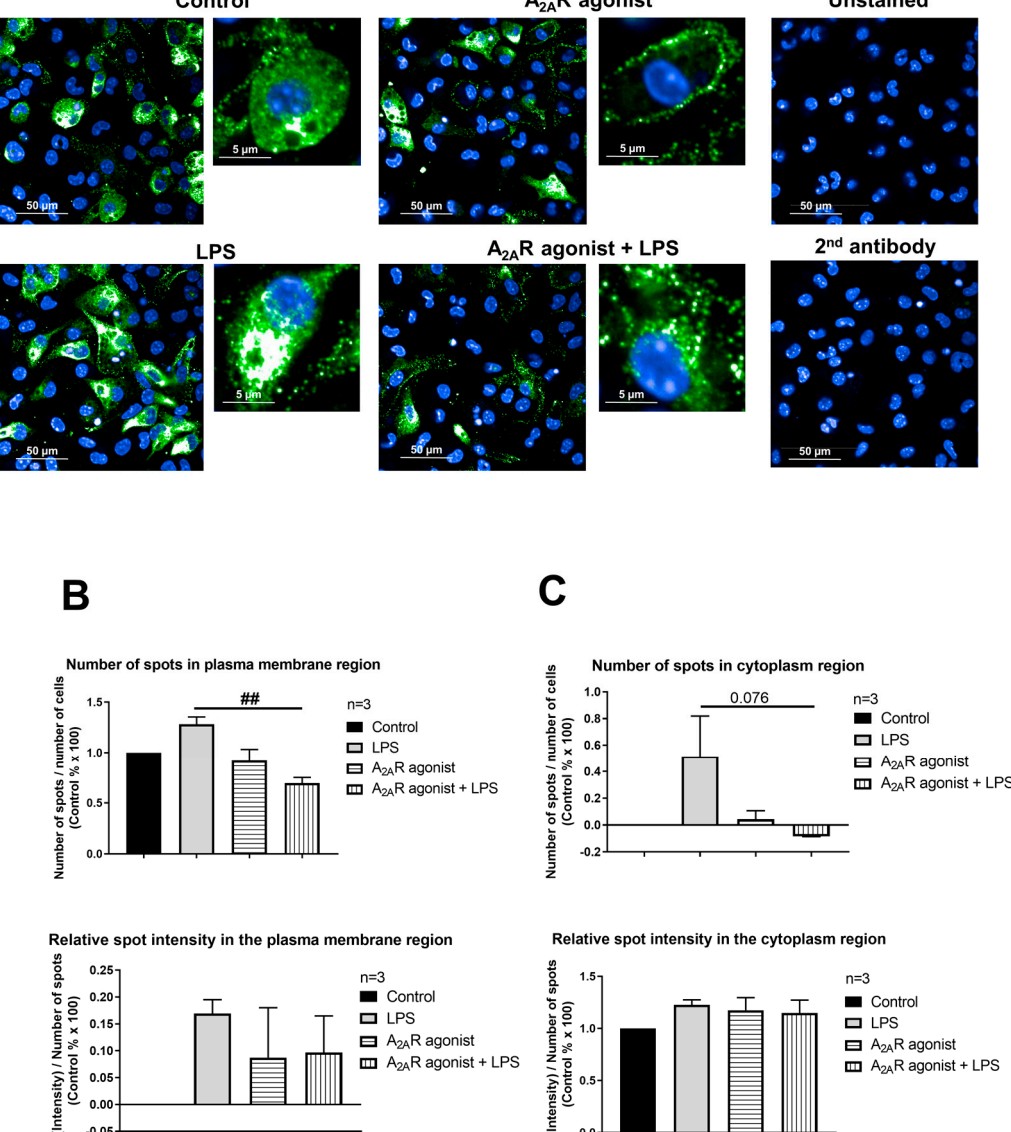

**Figure 8.** $A_{2A}R$ activation decreases EEA1 expression in mouse IPMφs. (**A**) Immunofluorescence staining of IPMφ cells was made using EEA1 specific (SAB4300682 Sigma-Aldrich, St. Louis, MO, USA) primary and Alexa-488 conjugated anti-rabbit secondary antibody (A27034, Thermo Fisher Scientific, Waltham, MA, USA) (green). Nuclei of macrophages were stained with DAPI (blue) (D1306, Thermo Fisher Scientific, Waltham, MA, USA). EEA1 specific fluorescence intensity was measured after LPS activation and treatment with the $A_{2A}R$ agonist CGS21680 by Opera Phenix High Content Confocal System (Perkin Elmer, Waltham, MA, USA). Fifty fields and 370–3445 cells were acquired per well, and laser-based autofocus was performed at each imaging position. Images of DAPI and Alexa-488 channels were collected at 2 μm of the Z image plane using a 63× water immersion objective (NA: 1.15). Cellular features, such as the number of spots and relative spot intensities in the (**B**) membrane and (**C**) cytoplasmic regions, were extracted. Data obtained from the individual analysis of 370–3445 different cells are presented as mean ± SEM. ## $p < 0.01$ LPS vs. LPS + $A_{2A}R$ agonist-treated cells.

## 4. Discussion

In the present study, we report that $A_{2A}R$ interacts with the NPC1 protein in macrophages. To our knowledge, this is the first-time demonstration of an interaction between the NPC1 protein and a G protein-coupled receptor (GPCR). The interaction was identified by two independent proteomic approaches using the C-terminus of $A_{2A}R$ as bait, thus yielding 27 coincidence interactors. The molecular interaction of $A_{2A}R$ and NPC1 protein was further validated in HEK-293 cells permanently expressing $A_{2A}R$ and RAW264.7 cells endogenously expressing the receptor (Figure 2). Interestingly, stimulation of $A_{2A}R$ reduced the expression of NPC1 mRNA, the density of the NPC1 protein, and its targeting of the cell surface in macrophages. Importantly, this effect was dependent on the activation status of the macrophages. Furthermore, stimulation of $A_{2A}R$ also altered the density of LAMP2 and EEA1, two endosomal markers associated with the NPC1 protein, which are also involved in endocytic trafficking and regulation of intracellular cholesterol transport.

The detailed Investigation of the interaction between $A_{2A}R$ and NPC1 was motivated by the observation that NPC1 has been previously shown to be functionally related to the $A_{2A}R$-mediated signaling pathway. Popoli's group has shown that $A_{2A}R$ activation restores mitochondrial membrane potential and cholesterol accumulation in fibroblasts from NPC1 patients and in human neuronal and oligodendroglial cell lines [36,37]. Furthermore, it has been described that receptor activation significantly reduces the defect in the intracellular transport of endocytosed cholesterol in NPC1-deficient fibroblasts and oligodendrocytes [36,45]. Macrophages and macrophage-like cells, such as dendritic cells, microglia, and osteoclasts, also rely on vesicular trafficking for fighting infections, general housekeeping functions and for tumors. $A_{2A}R$-mediated signaling plays an important role in the self-renewal of macrophages in the tumor microenvironment [46].

Grinstein and co-workers have already shown in RAW264.7 macrophage cells and fibroblasts that the NPC1 protein plays a role in the intracellular accumulation of cholesterol [47].

Collectively, these findings suggest that the interaction between $A_{2A}R$ and the NPC1 protein is biologically relevant and that $A_{2A}R$ may play a role in regulating the function of the NPC1 protein in macrophages. One possibility for the interaction of $A_{2A}R$ and NPC1 proteins is when lysosomal proteins are transported into the cytoplasm due to permeabilization or damage of the lysosomal membrane (Reviewed in [48]), and NPC1 may then contact the C-terminal domain of $A_{2A}R$. Another possibility is when receptors and their ligands can be internalized from the cell surface by endocytosis. The best-known mechanism of endocytosis is clathrin-mediated endocytosis, whereby transmembrane receptors and their bound ligands localize to specific membrane microdomains. These so-called clathrin-coated vesicles fuse with early endosomes, and receptor-ligand complexes are transported to different cellular compartments, such as multivesicular bodies [49]. During this process, the intracellular domain of $A_{2A}R$ may also associate with the lysosomal NPC1 protein.

Bernardo and colleagues are investigating myelin defects and the role of cholesterol in myelination. Elevation of adenosine levels and stimulation of $A_{2A}R$ may offer a therapeutic perspective in NPC, as it has a beneficial effect on the dysmyelination [50]. Previously, Npc1 gene expression was observed to be affected by cycloheximide and progesterone. Cycloheximide increases Npc1 mRNA levels. Granulosa cells may be subjected to transient progesterone-induced Npc1 blockade, resulting in a compensatory increase in Npc1 mRNA and NPC1 protein. By stabilizing Npc1 mRNA, progesterone prepares luteinizing granulosa cells for increased LDL flux through the endosome/lysosome compartments [51]. We also examined how $A_{2A}R$ activation affects NPC1 mRNA and protein expression in IPMϕ cells and found that $A_{2A}R$ agonist treatment at both molecular levels reduced NPC1 expression in LPS-activated macrophages.

To determine whether $A_{2A}R$ activation directly affects NPC1 protein abundance, changes in protein expression and localization were examined by high-content confocal microscopy, while NPC1 was detected with the specific antibody. The results showed that $A_{2A}R$ agonist treatment reduced both the number of NPC1-specific spots and the total spots area in the plasma membrane and cytoplasmic regions of macrophages compared to LPS-activated samples (Figures 4, 5 and S1).

$A_{2A}R$ signaling has been linked to the regulation of vesicular trafficking. For example, Isidoro and colleagues (2004) found that pretreatment with an $A_{2A}R$ agonist-induced the movement of endosomes and lysosomes towards the plasma membrane, followed by fusion of these organelles with the plasma membrane. This was evidenced by the appearance of lysosome-associated membrane protein 1 (LAMP1) on the cell surface and the release of lysosomal soluble enzymes by hepatocytes [52]. Since NPC1 is a protein located in the inner membrane of lysosomes and its function is to transport LDL-derived cholesterol from the lumen of the lysosome to the membrane [42], we investigated changes in the expression and localization of another lysosomal marker, the LAMP2 protein [53] following treatment with the $A_{2A}R$ agonist (CGS21680) in macrophage cells. We demonstrated that $A_{2A}R$ activation significantly reduced the amount of LAMP2 protein in the membrane and cytoplasmic regions of RAW 264.7 (Figure 6B) and IPMφ cells (Figure 7B), compared to the change upon LPS activation. Our results were in agreement with observation demonstrating that $A_{2A}R$ activation leads to decreased LAMP2 expression in fibroblasts from both healthy and NPC1 patients [37].

Tahirovic and Hecimovic and their workgroups observed enlarged early endosomes and recycling endocytic compartments in CHO cells from NPC1 KO mice [44]. This suggested that the absence of NPC1 protein affects endocytic organelle function through defects in the endolysosomal pathway. By testing the EEA1 marker, which has been successfully used previously in the macrophages [43], we tested how $A_{2A}R$ activation alters EEA1 protein abundance and intracellular localization. Similar to LAMP2, EEA1 protein levels showed a significant decrease after $A_{2A}R$ agonist treatment in the membrane and cytoplasmic regions of peritoneal macrophages compared to the increased levels after LPS activation (Figures 8B and S4).

Vesicular transport is a critical cellular process that facilitates the movement of molecules between specific membrane-bound compartments. Transport vesicles, which are formed in the donor compartment, transfer cargo to acceptor sites along various pathways, including the biosynthetic and endocytic routes. These vesicles play a crucial role in cellular trafficking and help maintain proper organelle function and homeostasis. In our proteomic approach, we identified 27 $A_{2A}R$ interactors in macrophages. Functional analysis of the proteins identified by two independent methods revealed that eight of them play a role in cellular vesicular trafficking (highlighted in blue in Table 3). These are Coatomer subunit gamma-2 (COPG2), Niemann–Pick C1 protein (NPC1), Sec1 family domain-containing protein 1 (SCFD1), Adaptor Related Protein Complex 3 Subunit Mu-1 (AP3M1), Ras-related protein Rab-18 (RAB18), RUN and FYVE domain-containing protein (RUFY1), Receptor-mediated endocytosis-8 (RME-8)/DnaJ Heat Shock Protein Family (Hsp40) Member C13 (DNAJC13)/, Annexin A5 (ANXA5). The interactions formed by $A_{2A}R$ with proteins involved in vesicular trafficking further strengthen the regulatory role of $A_{2A}R$ in this process.

Our result contributes to a more detailed understanding of the molecular mechanism of NPC1 disease by characterizing the functional analysis of NPC1 and $A_{2A}R$ (Figure 9).

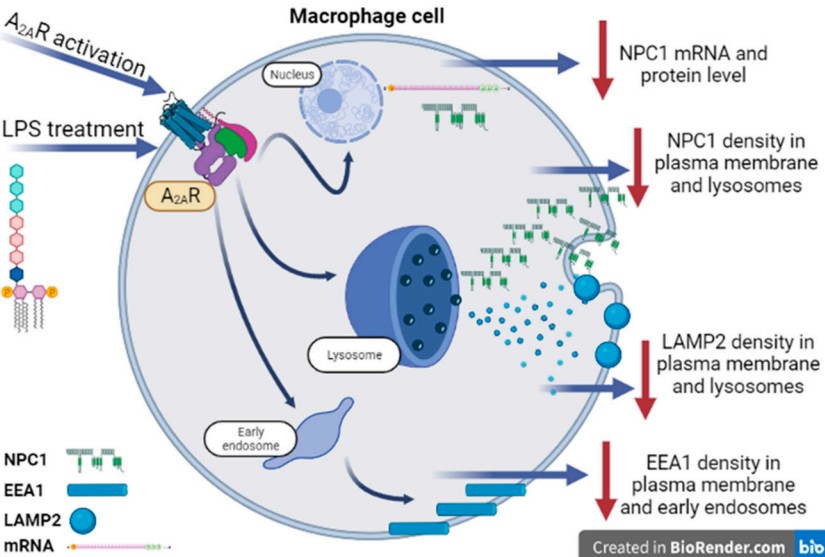

**Figure 9.** Proposed role of $A_{2A}$R-NPC1 protein interaction in activated macrophages. The results of our experiments on a large number of cells showed that $A_{2A}$R agonist treatment reduces NPC1 mRNA and protein levels in LPS-activated macrophages in a similar manner to the applied LAMP2 and EEA1 markers.

**Supplementary Materials:** The following supporting information can be downloaded at: https://www.mdpi.com/article/10.3390/cimb45060315/s1 (Figures S1–S6) and can be found at the Department of Medical Chemistry http://193.6.152.202:5000/ accessed on 24 May 2023.

**Author Contributions:** Study conception and design: E.K. and G.H.; Project supervision: E.K.; Data collection: A.S. and C.B.; Data analysis and interpretation: A.S., G.U., A.T.G., Z.B., F.C. and E.K.; Drafting: G.H., L.V., F.C. and E.K. All authors have read and agreed to the published version of the manuscript.

**Funding:** EK received funding from National Research Development and Innovation Office OTKA MB08A 84685; LV received funding from the National Research, Development and Innovation Office grants GINOP-2.3.2-15-2016-00020 TUMORDNS, GINOP-2.3.2-15-2016-00048-STAYALIVE and National Research Development and Innovation Office OTKA K132193, K147482. GH received funding from National Institutes of Health grants R01GM066189, R01DK113790 and R01HL158519.

**Institutional Review Board Statement:** All animal experiments were conducted according to the guidelines of the Declaration of Helsinki, and the animal study protocol was approved by the Institutional Review Board of the University of Debrecen (DEMÁB 15/2016).

**Informed Consent Statement:** Not applicable.

**Data Availability Statement:** Data supporting reported results can be found at the Department of Medical Chemistry http://193.6.152.202:5000/ accessed on 24 May 2023.

**Acknowledgments:** The authors thank Tankáné Andrea Farkas for technical assistance (Department of Medical Chemistry, Faculty of Medicine, University of Debrecen, H-4032 Debrecen, Hungary), Eszter Janka for the revision of the statistical analysis (Department of Dermatology, Faculty of Medicine, University of Debrecen, H-4032 Debrecen, Hungary).

**Conflicts of Interest:** The authors declare no conflict of interest.

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
