# Peer review of "Adenosine A2A Receptor Activation Regulates Niemann–Pick C1 Expression and Localization in Macrophages"

_cimb, doi:10.3390/cimb45060315_

Round 1

Reviewer 1 Report (Previous Reviewer 2)

In this manuscript, the authors investigated the interaction between A2AR and the NPC1 protein. By using two independent proteomic approaches, they found that full-length A2AR interact with NPC1 in HEK293 and macrophages. In addition, they observed that A2AR signaling is involved in the regulation of LAMP2 and EEA1 expression in macrophages.

Overall, the results presented in this manuscript are of interest. The plan experiment is well designed.

Question:

1)     It there any evidence about the functional interaction between A2AR and NPC1 in tumor-associated macrophages (TAMs)?

English editing looks fine

Author Response

Answers to Reviewer 1

Title

Adenosine A2A receptor activation regulates Niemann-Pick C1 expression and localization in macrophages

Authors

Adrienn Skopál , Gyula Ujlaki , Attila Tibor Gerencsér , Csaba Bankó , Zsolt Bacsó , Francisco Ciruela , László Virág , György Haskó , Endre Kókai *

     We thank the Editor for the judicious handling of the manuscript. Thus, based on the encouraging opinion and the comments provided by the reviewers, we have prepared a new version of the manuscript considering all the suggestions made. Accordingly, we are submitting a new version of the manuscript that has been improved thanks to the reviewers’ comments.

     Please, find attached or point-by-point answer to reviewer’s concerns:

Answer for Reviewer 1

In this manuscript, the authors investigated the interaction between A2AR and the NPC1 protein. By using two independent proteomic approaches, they found that full-length A2AR interact with NPC1 in HEK293 and macrophages. In addition, they observed that A2AR signaling is involved in the regulation of LAMP2 and EEA1 expression in macrophages.

Overall, the results presented in this manuscript are of interest. The plan experiment is well designed.

Question:

1) Is there any evidence about the functional interaction between A2AR and NPC1 in tumor-associated macrophages (TAMs)?

Our results suggest that increased expression and activation of A2AR in activated macrophages reduces NPC1 protein expression. This observation is similar to those previously described for NPC2, in that NPC2 expression is reduced in human hepatocellular carcinoma cells and promotes tumorigenesis. Furthermore, in addition to NPC2, the expression of Niemann-Pick C1-Like 1 (NPC1L1), a homologous protein of NPC1 (Chen et al. 2018; PMID: 36636588), was also downregulated, which promoted tumor progression.

In addition, it was also observed that the culture medium of hepatocarcinoma cells enhances macrophage A2AR expression, resulting in increased macrophage proliferation (Filiberti et al. 2022; PMID: 36359228).

These observations provide further evidence for a functional link between A2AR and NPC1 that may be important in the regulation of polarization of TAMs.

Reviewer 2 Report (New Reviewer)

The authors studied the interactions between A2A receptor and Npc1 protein in different cells, including macrophages and found out that activation of A2A negatively regulates the increased expressions of Npc1 in both cellular membrane and cytoplasma upon LPS stimulation in macrophages. Generally, these results are interested for us to understand the functional regulation of Npc1. Still some points should be addressed.

Major point:

In Figs.4-7, the membrane expression of the proteins, like Npc1, LAMP2, and EEA1, was difficult to recognise and the authors should do double-immunostaining using the membrane protein marker to identify the location of these proteins and show them by a high magnification as Figure S2.

Minor point:

A short introduction of Npc1 and NPC should be given in the Introduction part, but not in the Discussion part.

Figure resolution should be higher with a large magnification.

Round 2

Reviewer 2 Report (New Reviewer)

My points were addressed.

This manuscript is a resubmission of an earlier submission. The following is a list of the peer review reports and author responses from that submission.

Round 1

Reviewer 1 Report

In their manuscript, Skopal and co-workers applied proteomic approaches and identified adenosine A2A receptor (A2AR) to interact with the endolysosomal cholesterol transporter Niemann Pick type C1 (NPC1) via the C-terminal receptor tail. The results presented in the first part dealing with the initial biochemical characterization of A2AR-NPC1 interaction were carried out in a murine macrophage cell line ectopically expressing a myc-tagged receptor, complemented by GST-pulldown experiments using a GST-A2AR tail fusion construct and cell lysates. Samples were analyzed by mass spectrometry. The Coomassie-stained SDS-PAGEs in Fig. 1 don’t support the results as they don’t show any specific band that was pulled down and can therefore be omitted. The verification that the endogenous receptor can interact with NPC1 and Western blots for NPC1 that are necessary to support the conclusion of a specific interaction are presented in Fig.2. However, appropriate controls are missing (no irrelevant antibody was included and controls were done in the absence of anti-A2AR antibody).

The second part of the results aims to link the findings with a physiological relevance of this interaction. Whereas A2AR activation in peritoneal macrophages did not change harvested from LPS-treated mice did not change NPC1 mRNA levels, the decrease induced by LPS was more pronounced when cells were additionally stimulated with the A2AR agonist (calculation of the changes is not clear to me, because the legend states that levels were compared to LPS-treated cells, however, a change in LPS-treated cells is presented, what are the controls then?). This was more or less also seen on the protein levels quantitatively, although the representative western blot in Fig. 3B does not reflect the quantitative data (a weaker band in LPS-treated cells and almost identical densities in A2AR-activated cells).  Fig. 4 presents an immunofluorescence-based evaluation of the intracellular localization and intensity of NPC1 in these cells. However, the results do not support the statement that „A2AR stimulation decreases the presence of NPC1 protein in both the plasma membrane and cytoplasmic region of LPS-activated macrophages“ (lines 206, 206) as the spot intensities are increased in both cytosol and plasma membrane when compared to LPS treatment alone. Moreover, these results only suggest an NPC1 translocation upon activation of the cells either via LPS or A2AR agonist and do not support any claims that the A2AR-NPC1 interaction is relevant for these observations. The same holds for Figs 6,7, and 8.  

Overall, the initial finding of a specific A2AR interaction with NPC1 based on biochemical analysis is not supported by experimental data strong enough to support the conclusion. The data on functional results of the interaction elucidate an interplay between LPS and A2AR activation but are not dealing with A2AR-NPC1 interaction. In this regard, the title is misleading and does not reflect the data presented in the manuscript. In the discussion section, the authors do not provide a model of how the interaction might take place. NPC1 is located on the luminal membranes of endolysosomes and A2AR is a receptor on the plasma membrane with the C-terminal tail oriented toward the cytosol, where will this interaction happen?  

Author Response

Answers to Reviewer 1

Title

Adenosine A2A receptor activation regulates Niemann-Pick C1 expression and localization in macrophages

Authors

Adrienn Skopál , Gyula Ujlaki , Attila Tibor Gerencsér , Csaba Bankó , Zsolt Bacsó , Francisco Ciruela , László Virág , György Haskó , Endre Kókai *

     We thank the Editor for the judicious handling of the manuscript. Thus, based on the encouraging opinion and the comments provided by the reviewers, we have prepared a new version of the manuscript considering all the suggestions made. Accordingly, we are submitting a new version of the manuscript that has been largely improved thanks to the reviewers’ comments.

     Following the instructions of the reviewer 1, we have double checked the title of the manuscript, thus we believe that it meets the criterion of being understandable to non-expert readers in the field.

     Please, find attached or point-by-point answer to reviewer’s concerns:

Reviewer 1

Comments and Suggestions for Authors

In their manuscript, Skopal and co-workers applied proteomic approaches and identified adenosine A2A receptor (A2AR) to interact with the endolysosomal cholesterol transporter Niemann Pick type C1 (NPC1) via the C-terminal receptor tail. The results presented in the first part dealing with the initial biochemical characterization of A2AR-NPC1 interaction were carried out in a murine macrophage cell line ectopically expressing a myc-tagged receptor, complemented by GST-pulldown experiments using a GST-A2AR tail fusion construct and cell lysates. Samples were analyzed by mass spectrometry.

The Coomassie-stained SDS-PAGEs in Fig. 1 don’t support the results as they don’t show any specific band that was pulled down and can therefore be omitted.

Answer:

Figure 1 shows an image of the gels sent for mass spectrometric analysis on panels C and D. The sensitivity of the Coomassie Brilliant Blue G250 staining does not always allow detection of proteins associated with the "bait" protein by GST-pulldown. In panel C of Figure 1, a large protein band (above 180 kDa molecular mass), which is assumed to be the NPC1 protein based on its size, can be detected in the cMyc antibody specific complex. At the request of the reviewer, panel D of Figure 1 is removed.

The verification that the endogenous receptor can interact with NPC1 and Western blots for NPC1 that are necessary to support the conclusion of a specific interaction are presented in Fig.2. However, appropriate controls are missing (no irrelevant antibody was included and controls were done in the absence of anti-A2AR antibody).

Answer:

The immunoprecipitation Western blotting experiment shown in panel D of Figure 2 was also performed with a non-specific rabbit control serum in addition to the anti-A2AR specific antibody, as suggested by the reviewer. The result of the experiment confirmed that NPC1 protein was only detectable in the immunocomplex in the presence of the anti-A2AR specific antibody, no NPC1 specific signal was obtained when using the control serum.

The second part of the results aims to link the findings with a physiological relevance of this interaction.

Whereas A2AR activation in peritoneal macrophages did not change harvested from LPS-treated mice did not change NPC1 mRNA levels, the decrease induced by LPS was more pronounced when cells were additionally stimulated with the A2AR agonist (calculation of the changes is not clear to me, because the legend states that levels were compared to LPS-treated cells, however, a change in LPS-treated cells is presented, what are the controls then?).

Answer:

The change in NPC1 mRNA abundance upon LPS treatment was compared to the NPC1 mRNA abundance in untreated peritoneal macrophages. We found that LPS treatment significantly decreased the amount of NPC1 mRNA. Furthermore, we compared LPS-treated and LPS+A2AR agonist-treated macrophages and showed that co-treatment (LPS+A2AR agonist) caused a further significant reduction in NPC1 mRNA compared to LPS-treated samples alone. In panel A of Figure 3, the Y-axis label was changed from "Fold change" to "Relative amount of NPC1 mRNA" because the data were transformed for one-way ANOVA and supplemented with Sidak's post-hoc test in the statistical analysis. The results are plotted on the graph, where the value of the untreated control is zero.

This was more or less also seen on the protein levels quantitatively, although the representative western blot in Fig. 3B does not reflect the quantitative data (a weaker band in LPS-treated cells and almost identical densities in A2AR-activated cells).

Answer:

In Figure 3, we replaced the result showing NPC1 protein and actin expression with a more representative WB images. The change in the expression of NPC1 protein was normalized to the intensity of the bands provided by the actin protein in the WB experiment shown in Figure 3. panel B. The densitometry analysis was based on four independent experiments.

Fig. 4 presents an immunofluorescence-based evaluation of the intracellular localization and intensity of NPC1 in these cells. However, the results do not support the statement that „A2AR stimulation decreases the presence of NPC1 protein in both the plasma membrane and cytoplasmic region of LPS-activated macrophages“ (lines 206, 206) as the spot intensities are increased in both cytosol and plasma membrane when compared to LPS treatment alone.

Answer:

Yes, the critic is right. In the case of the macrophage cell line RAW264.7, we cannot state that "A2AR stimulation decreases the presence of NPC1 protein in both the plasma membrane and cytoplasmic region of LPS-activated macrophages". The conclusion was modified as follow:

„A2AR stimulation decreases the presence of NPC1 protein in the plasma membrane of LPS-activated macrophages“

Moreover, these results only suggest an NPC1 translocation upon activation of the cells either via LPS or A2AR agonist and do not support any claims that the A2AR-NPC1 interaction is relevant for these observations. The same holds for Figs 6, 7, and 8.

Answer:

We aim to demonstrate the functional relationship between A2AR and NPC1 in macrophage cells by changes in the amount and localization of NPC1 protein in response to LPS activation and A2AR agonist treatment. However, this functional relationship was brought to our attention by the results of our biochemical studies. The clarification made by the reviewer is taken into account in the Discussion.

Overall, the initial finding of a specific A2AR interaction with NPC1 based on biochemical analysis is not supported by experimental data strong enough to support the conclusion. The data on functional results of the interaction elucidate an interplay between LPS and A2AR activation but are not dealing with A2AR-NPC1 interaction. In this regard, the title is misleading and does not reflect the data presented in the manuscript.

The title of the manuscript has been changed.

The new title is:

Adenosine A2A receptor activation regulates Niemann-Pick C1 expression and localization in macrophages”

In the discussion section, the authors do not provide a model of how the interaction might take place. NPC1 is located on the luminal membranes of endolysosomes and A2AR is a receptor on the plasma membrane with the C-terminal tail oriented toward the cytosol, where will this interaction happen?

Answer:

One possibility for the interaction of A2AR and NPC1 proteins is when lysosomal proteins are transported into the cytoplasm due to lysosomal membrane permeabilization or damage (Reviewed in Wang F. et al. 2018) and can contact the C-terminal domain of A2AR.

The other possibility for the interaction when receptors and their ligands can be internalized from the cell surface by several mechanisms of endocytosis. The best-known mechanism of endocytosis is clathrin-mediated endocytosis, whereby transmembrane receptors and their bound ligands are localized to specific membrane microdomains. These so-called clathrin-coated vesicles fuse with early endosomes and the receptor-ligand complexes are transported to different cellular compartments, such as multivesicular bodies (Tchikov V. et al. 2011). During this process, the intracellular domain of A2AR can come into contact with the lysosomal NPC1 protein.

Wang F., Gómez-Sintesatricia R., Boya P. Lysosomal membrane permeabilization and cell death Wiley Traffic. 2018;19:918–931.

Tchikov V., Bertsch U., Fritsch J., Edelmann B., Schütze S. Subcellular compartmentalization of TNF receptor-1 and CD95 signaling pathways European Journal of Cell Biology 90 (2011) 467–47

Reviewer 2 Report

In this manuscript, Adrienn Skopál and collaborators investigated the interaction between A2AR and the Niemann-Pick type C intracellular cholesterol transporter 1 (NPC1) protein. By using two independent proteomic approaches, they found  that  full-length A2AR interact with NPC1 in HEK293 and macrophages. In addition, they observed that A2AR signaling in involved in the regulation of LAMP2 and EEA1 expression in macrophages.

Overall, the results presented in this manuscript are just descriptive and too preliminary. More experiments are required to determine the molecular mechanism underlying the role of A2AR on NPC1, LAMP2 and EEA1 expression.  

Questions:

 1)     Fig 4-8: There is no evidence that A2AR is activated in macrophages after incubation with A2AR agonist. In addition, which is the effect of LPS on A2AR expression and signaling cascade?

2)     Fig 4: the quality and the magnification of the images are not sufficient to quantify the difference of NPC1 spots in cytoplasm and membrane. Are the authors sure that cell images were acquired using 63x?

3)     References should be updated: For example, A2AR in involved in tumor-associated macrophages (TAMs) renewal (PMID: 36359228).

4)     Did the authors tested the functional interaction between A2AR and NPC1 in TAMs?

Author Response

Answers for Reviewer 2

Title

Adenosine A2A receptor activation regulates Niemann-Pick C1 expression and localization in macrophages

Authors

Adrienn Skopál , Gyula Ujlaki , Attila Tibor Gerencsér , Csaba Bankó , Zsolt Bacsó , Francisco Ciruela , László Virág , György Haskó , Endre Kókai *

     We thank the Editor for the judicious handling of the manuscript. Thus, based on the encouraging opinion and the comments provided by the reviewers, we have prepared a new version of the manuscript considering all the suggestions made. Accordingly, we are submitting a new version of the manuscript that has been largely improved thanks to the reviewers’ comments.

     Following the instructions of the reviewer 1, we have double checked the title of the manuscript, thus we believe that it meets the criterion of being understandable to non-expert readers in the field.

     Please, find attached or point-by-point answer to reviewer’s concerns:

Reviewer 2

Comments and Suggestions for Authors

In this manuscript, Adrienn Skopál and collaborators investigated the interaction between A2AR and the Niemann-Pick type C intracellular cholesterol transporter 1 (NPC1) protein. By using two independent proteomic approaches, they found that full-length A2AR interact with NPC1 in HEK293 and macrophages. In addition, they observed that A2AR signaling in involved in the regulation of LAMP2 and EEA1 expression in macrophages.

Overall, the results presented in this manuscript are just descriptive and too preliminary. More experiments are required to determine the molecular mechanism underlying the role of A2AR on NPC1, LAMP2 and EEA1 expression.  

Questions:

1) Fig 4-8: There is no evidence that A2AR is activated in macrophages after incubation with A2AR agonist. In addition, which is the effect of LPS on A2AR expression and signaling cascade?

Answer:

Before performing the experiments, we tested the effect of LPS activation and CGS 21680, as an A2AR agonist on A2AR activation. To verify the effect of CGS 21680 and LPS treatment, we examined the changes in mRNA expression of inflammatory cytokines (TNF-a and IL-6) regulated by receptor activation. After 4 h of LPS (100 ng/ml) activation, mRNA levels of both cytokines tested were significantly increased in both RAW 264.7 and peritoneal macrophages, and increased inflammatory cytokine mRNA expression was significantly reduced by administration of the A2AR agonist (100 nM) in activated macrophages.

The effect of LPS activation on A2AR expression and its regulated signalling was shown in our previous work (Skopál et al. 2022) and we found that LPS treatment (100 ng/ml) enhances A2AR expression in peritoneal macrophages.

2) Fig 4: the quality and the magnification of the images are not sufficient to quantify the difference of NPC1 spots in cytoplasm and membrane. Are the authors sure that cell images were acquired using 63x?

Answer:

Images were taken with a high throughput confocal microscope with a 63x water immersion objective. The resolution of these images allowed the examination of the fluorescence intensity of the NPC1, LAMP2, EEA1 protein specific "spots". In the version of the manuscript submitted for peer review, the microscopic images show a full field of view area from these images. The quality and resolution of the images allow a smaller area to be shown at higher magnification, representing changes in the amount and localization of the proteins being studied. As requested by the reviewer, these recordings are presented in a corrected version of the manuscript.

3) References should be updated: For example, A2AR in involved in tumor-associated macrophages (TAMs) renewal (PMID: 36359228).

Answer:

The reference suggested by the reviewer will be included in the discussion.

4) Did the authors tested the functional interaction between A2AR and NPC1 in TAMs?

Answer:

The functional relationship of A2AR-NPC1 in TAM has not yet been investigated.

Skopal, A., T. Keki, P.A. Toth, B. Csoka, B. Koscso, Z.H. Nemeth, L. Antonioli, A. Ivessa, F. Ciruela, L. Virag, G. Hasko, and E. Kokai, Cathepsin D interacts with adenosine A(2A) receptors in mouse macrophages to modulate cell surface localization and inflammatory signaling. J Biol Chem, 2022. 298(5): p. 101888.

Round 2

Reviewer 2 Report

The new version of the manuscript is sufficiently improved.